# Electrogeneration and Characterization of Poly(methylene blue) Thin Films on Stainless Steel 316 Electrodes—Effect of pH

**DOI:** 10.3390/molecules29163752

**Published:** 2024-08-07

**Authors:** José Juan García-Jareño, Jerónimo Agrisuelas, Zoe Vargas, Francisco Vicente

**Affiliations:** Departament Química-Física, Universitat de València, Dr. Moliner 50, 46100 Burjassot, Spain; jeronimo.agrisuelas@uv.es (J.A.); zovarve@alumni.uv.es (Z.V.); francisco.vicente@uv.es (F.V.)

**Keywords:** methylene blue, stainless steel 316, polymerization, pH dependence, kinetics mechanism, digital video electrochemistry (DVEC)

## Abstract

Methylene blue was electropolymerized on the surface of stainless steel 316. The addition of sodium oxalate and working at a pH near 11 allowed us to obtain steel electrodes coated with an electroactive polymer. This polymer shows electrochromic properties like those of the monomer, but also exhibits electroactivity at more positive potentials, which is associated with the active centers in the bridges between monomeric units. A digital video electrochemistry study allowed us to simultaneously quantify, on the one hand, the color changes on the entire surface of the stainless steel and on the other to separate the contribution of the active centers of the phenothiazine ring and of the inter-monomer bonds to the overall polymer response by means of assessing the mean color intensities. A reduction mechanism for the polymer, compatible with the pH variation of the observed electrochemical behavior, was proposed.

## 1. Introduction

Conductive polymers (CPs) have emerged as an attractive class of materials, gaining considerable interest for their conductive properties. This has now made it possible to use them to replace some metals or semiconductors of an inorganic nature in certain applications [1]. The first paper on this subject was the discovery of polyacetylene (PA) by Hideki Shirakawa, Alan MacDiarmid and Alan Heeger [2], from which numerous other conductive polymers with similar properties, such as polyphenylene (PP) [3], polyphenylenevinylene (PPV) [4], polypyrrole (PPy) [5,6], (PTh) [7] and polyaniline (PANI) [8,9], were synthesized. Nowadays, a large number of different monomers are known to form conductive polymers [4,10].

A huge challenge lies in achieving precise control of the electrical or electrochemical properties of CPs. Despite their advantages, like their chemical diversity, low density, flexibility, corrosion resistance and the ability to easily control their shape and morphology, as well as their adjustable conductivity compared to other inorganic equivalents, they still have inherent limitations in terms of solubility, conductivity and long-term stability. Consequently, research has been driven towards the creation of CP-based complexes and hybrids with innovative structures, leading to the development of materials with high mechanical stability, flexibility and conductivity. This has significantly expanded their applications as key components in light-emitting diodes, transistors, electronic actuators and other devices [1].

CP synthesis usually involves the oxidation of the monomer by oxidizing agents or the application of electric potential or current through electrodes [11]. Thanks to the system of alternating single (σ) and double (π) bonds present in CPs, these π conjugated systems could acquire electrically conductive properties by losing electrons through oxidation or by receiving electrons through reduction. Therefore, the oxidation of CPs generates a p-type doped state and reduction generates an n-type doped state [12]. This means that conductivity of CPs can also be controlled by these redox reactions [11]. As in non-conductive polymers, the general physical properties of CPs depend on their size, length and molecular weight [11], which, in the case of conductive polymers, can be modulated by the electrogeneration of experimental conditions.

Among these types of conductive polymers, poly(phenothiazines) and poly(phenazines) [13,14] are worth mentioning. These monomers are characterized by a central aromatic group and two more or less methylated amino groups at the two ends. These monomers have been used as dyes since they show intense absorption bands at wavelengths in the visible spectrum. It is known that after polymerization, this polymer retains a color similar to that of the monomer, which suggests that the chromophore group is not highly affected after polymerization. For example, many poly(phenothiazines) have a characteristic blue color (poly(Azure A), poly(methylene blue), etc.). In the case of poly(phenazine) such as poly(Neutral Red), the characteristic red color of this polymer, as contrasted with the blues of the phenothiazines, confirms the idea that the chromophore group is the central aromatic group of the monomer. However, new absorption bands at wavelengths in the near-IR spectrum, which are associated with the bonds between monomeric units, have been reported to occur after polymerization [15]. Consequently, the use of spectroelectrochemical techniques may be the best option to characterize this type of materials and the mechanisms by which the oxidation and reduction processes take place.

One of these monomers, despite having been known for years, may gain attention in the coming years due to different properties that make it very interesting in different applications. Methylene blue (MB) (Figure 1) was first reported by the German chemist Heinrisch Caro in 1876; a few years later, in 1886, Dr. Paul Ehrlich proved that MB stains living neurons and also plasmodium, the parasite that causes malaria. It was later found to have properties that inhibit the transmission of malaria [16]. More recent research suggests that this pigment may also be useful in the treatment of neurodegenerative diseases or cancer. Even in the recent COVID-19 pandemic, it was found that it inhibits the interaction between the SARS-CoV-2 spike protein and angiotensin-converting enzyme 2 (ACE2), which is key in virus infection [17].

Another interesting feature of MB is its characteristic blue color, which absorbs radiation at wavelengths between 630 and 680 nm. This absorption can lead to the formation of reactive oxygen species (ROS), such as singlet oxygen ^1^$O_{2}$. It is precisely these ROS that can destroy cancer cells and control their growth. This means that this material can also be used in photodynamic therapies to control the proliferation of cancer cells or other diseases [18,19,20]. One of the key points in these electrochemical pollutant removal processes is precisely energy consumption. The possibility of generating ROS by the absorption of electromagnetic radiation can lead to a significant reduction in the consumption of electrical energy for normal operation. All this makes the optical properties of these materials of great interest for the design of these applications.

It is also important to keep in mind that for these materials to be used in some of these applications, it is necessary for MB to be fixed to some kind of support. In the case of medicinal applications to aid localized application and in the case of electrochemical applications, a good electronic conductor is necessary for it to be used as an electrode. The possibility of applying electromagnetic radiation to produce ROS would suggest that electrodes that do not absorb electromagnetic radiation but instead reflect it could significantly improve its efficiency. If we add to all this the fact that the substrate has good mechanical properties, it is not particularly expensive and it is easy to recover, stainless steel 316 (SS316) is a perfect candidate as a substrate to fix MB. Type 316 stainless steel is an austenitic chromium nickel stainless steel containing molybdenum. This addition increases its overall corrosion resistance, improves its resistance to the pitting of chloride ion solutions and provides increased strength at elevated temperatures.

Another problem with the use of MB is that it must be firmly attached to the substrate and not solubilized in aqueous media where it is to be employed. Simple adsorption on the substrate surface may not be sufficient; therefore, a good strategy is to electropolymerize the MB on SS316. Once polymerized, it will be necessary to check that the polymer retains the same optical and electrochemical properties of the monomer that make it interesting for applications that require the production of controlled ROS or for anion–cation insertion.

However, the steel also has some drawbacks to the generation of polymers on its surface. On SS316 electrodes, at the potentials at which the radical cation is generated, ROS and Fenton-like processes are also favored due to the presence of iron cations [21,22]. It is well known that the mineralization of MB can be achieved by Fenton-like processes in different experimental conditions [23,24,25] and if that happens, it will be impossible to obtain a good-quality polymer. This difficulty must be overcome before the polymer can be obtained on the surface of SS316.

As mentioned above, the optical–spectroscopic properties of these materials are very important for the accurate description of their oxidation or reduction mechanisms, since these involve color changes. Spectroelectrochemistry consists of linking spectroscopic techniques to electrochemical ones and thus simultaneously following changes in color or absorbance at different wavelengths and the electrical quantities associated with electrochemical experiment [5,26,27]. However, it requires the use of dark working conditions, and absorbance is measured only in a small region of the electrode surface. In an opaque electrode, as in our case, reflectance would have to be measured and the signal intensity would be quite dependent on the angle of incidence reflection. If the electrode surface did not reflect electromagnetic radiation, it would not be possible to follow color changes in these materials by spectrophotometric measurements. In recent years, a spectroelectrochemistry modality has been developed based on the acquisition of videos of the electrode surface during the experiment. The video frames are separated and treated as independent images. A region of the image is selected to follow its temporal evolution. For this purpose, the mean intensity of each color channel (Red, Green and Blue) and the standard deviation of each color intensity in the analyzed region are obtained. This technique is known as digital video electrochemistry (DVEC) and has been used to monitor the quality of metallic electrodeposits or conductive polymers, to obtain kinetic constants of electrochemical processes or to measure the rate and the extent of electrochemical processes in different parts of the electrode in a simulated manner [28,29,30,31,32].

The main objective of this research is essentially twofold. On the one hand, it is to optimize the electropolymerization of MB on SS316 to obtain poly(methylene blue) (PMB) as a thin film. As is well known, SS316 can oxidize under certain conditions and the surface of the metal will change its conductive properties, as there will be layers of metal oxides that can become insulating and therefore prevent its use as an electrode. The generation of the polymer requires the application of sufficiently positive potentials where the oxidation of the metal can occur. The first objective is to avoid this oxidation and other possible parallel reactions and ensure that only the oxidation of the monomer takes place so that the polymerization process can begin.

Secondly, once the polymer has been electrogenerated, it is necessary to verify that it retains the redox and chromatic properties of the monomer, in addition to verifying that the polymer is firmly fixed to the steel surface so that it can be used in different applications. For this purpose, its electrochromic response at different pH will be investigated to study the mechanism by which the redox processes of the polymer take place.

## 2. Results and Discussion

### 2.1. Electropolymerization on Stainless Steel 316

The first stage in this study is the generation of poly(methylene blue) on the SS316 surface. As mentioned in Section 1, the generation of the polymer requires the prior formation of cation radicals at sufficiently positive potentials. In the case of steels, at these potentials and depending on the medium, the oxidation of the metal can also take place, and at the end of the electrogeneration, there is no fixed polymer on the electrode surface.

As an example, Figure 2 shows that if the deposition is carried out under similar conditions to those performed on more stable electrodes such as gold, upon reaching sufficiently positive potentials, oxidation and pitting of the SS316 surface occurs, and although the surface apparently has a blue color, when this electrode is studied by cyclic voltammetry (CV) [33,34] in an aqueous medium, the blue color of the surface disappears and the solution is intensely blue-colored. This leads to the suspicion that the blue color on the surface is due to a weakly adsorbed monomer on the SS316 surface.

As discussed in Section 1, one of the compounds that is relatively easy to remove from water by Fenton-type advanced oxidation processes is methylene blue. Moreover, during the process of generating the cation radical, it is evident in Figure 2 how pitting corrosion of the steel has occurred and has released cations of different metals that are known to favor this type of advanced oxidation processes into the solution.

To try to overcome these problems, a different electropolymerization strategy is tested. On the one hand, in recent studies on polyaniline and poly(neutral red), it was proven that the presence of oxalate anions can stabilize the monomer, and then polymerization takes place [35,36] on SS316 electrodes. On the other hand, to try to avoid the passage of metallic cations into the solution, increases in pH are tested. An increase in pH causes the monomer units to deprotonate and decreases their solubility. In a recent study, it was proven that the maximum adsorption of methylene blue on a hydrogel is accomplished at pH 10.0 [37]; on the other hand, if the monomer is protonated, in some cases it may be more difficult to polymerize [38,39]. In addition, the properties of the electrogenerated polymer are dependent on the generation conditions, including the pH of the solution [40,41,42]. In addition, the chlorides that favor the formation of pitting are removed from the medium.

Figure 3 shows a polymer-coated SS316 electrode and how after the characterization study, the electrode shows a less intense blue color, although it still maintains a good electrochromic response.

After several trial-and-error tests, PMB is generated on the freshly polished electrode surface by performing up to 100 voltammetric cycles in the potential range from −0.4 V to 1.0 V against the reference electrode Ag|AgCl at a scan rate of 20 mV s^−1^. The monomer solution is brought to pH near 11 in the presence of oxalate anions (see Section 3).

Figure 4 shows how the current evolves during the polymer growth experiment. The region marked as (e) (potentials 0.8–1.0 V) is where the generation of the monomer cation radical and the polymer growth occurs. At more negative potentials, during the first few cycles, only a system of peaks marked (a) and (b) is observed for the anodic and cathodic peaks, respectively, which are associated with oxidation–reduction processes of the phenothiazine ring. As the number of cycles increases, a new system of voltammetric peaks appears (peaks (c) and (d), which were not observed during the first cycles). This system is associated with the electroactivity of the new bridges or bonds between monomeric units [13,15,30].

### 2.2. Spatiotemporal Digital Video Electrochemistry Analysis

The digital video electrochemistry technique allows for simultaneous and quantitative color changes to be obtained over the entire surface of the metal, provided that the solution is transparent. In the case of our SS316 coated with PMB and studied in a 0.5 M KNO_3_ solution by cyclic voltammetry, the surface changes to a lighter blue color when the polymer is reduced. Figure 5 shows a 3D map of color intensity increment ($\Delta I_{R},\;\Delta I_{G})$ or mean color intensity ($I_{R},\;I_{G})$ against the distance to the top of the electrode and the time elapsed during the voltammogram. One initial observation is that color changes take place simultaneously, regardless of the distance from the top of the electrode. In other systems, if there is a resistive electrode, color change and electrochemical processes take place on the electrode surface at different times [43,44,45]. Despite this homogeneity in terms of time, it is not the same if we look at the distance from the top of the electrode. Color intensity increment ($\Delta I_{C}$) proves more evident near the top of the electrode, while color intensity ($I_{C}$) changes prove larger at the bottom of the electrode. This discrepancy is explained by the fact that color increment is directly related to the number of active sites that have changed their oxidation state, while the color intensity change is explained because there are fewer active sites at the bottom.

In any case, the color changes do not occur as intensely across the whole electrode, possibly because the PMB has grown non-uniformly along the electrode. A larger amount of PMB could be generated on the top of the electrode, near the Teflon tape. If we want to correlate the color changes with the current during the voltammogram, we have to simultaneously analyze the entire electrode surface, since the measured current is the global current of the entire electrode surface. For this purpose, the best option is to obtain an average of the color intensity at each pixel for each of the three color coordinates (R, G, B). Note that this spatiotemporal analysis is not possible with a spectrophotometer that only takes the absorbance measurement at a localized region on the electrode surface.

### 2.3. Identification of Electroactive Sites in the Polymer—Deconvolution of Voltammetric Peaks 

Once a relatively stable polymer had been obtained on the SS316 electrode, it was time to characterize its electrochemical behavior. To achieve this, we used digital video electrochemistry (DVEC), which allowed us to simultaneously record the current signal during the electrochemical experiment and the video of the color changes that take place on the electrode surface [28,31,32,45,46]. DVEC was not possible for the electrogeneration experiment since the solution had a dark blue color and the electrode surface was masked by the solution. However, for the characterization study, all solutions were transparent, and it was relatively easy to capture images of the surface of the PMB-coated SS316 electrode.

The use of spectroelectrochemistry in this study aims to separate the different contributions to the electrochemical response in this type of materials [13,15,30]. As commented above, polymerization generates new active centers in the material when compared to the monomer in solution. Fundamentally, the new bridges between monomer units can exhibit electroactivity and also electrochromism [13,15].

First, a voltammogram with its associated color changes is presented to associate each color change with the electroactive processes (Figure 6).

Looking at the shape of the cathodic peak of electrical current (black line), it proves clear that this does not correspond to the shape of a voltammetric peak for adsorbed substances accordingly with the Laviron equation [47]. In some cases, the shape of the peaks is distorted by non-ideal effects such as resistance or interactions between active centers [43,44,48]. At other times, the peak simply does not correspond to a single electrochemical process and the measured current is the overlapping of different electrochemical processes [13,30,44]. In the latter case, coupled techniques such as quartz microbalance or spectroscopy can allow for the separation of the signal associated with different processes [49].

Figure 6 also includes color changes expressed as the derivative of color intensity versus time. A recent publication has shown that for voltammetric processes with associated color changes, current $i_{c}$ is directly related to the derivative of color intensity versus time $\;\left(\frac{dI_{C}}{dt}\right)$ [32,49].
(1)$$i_{C}=-\frac{nF}{k_{C}}\left(\frac{dI_{C}}{dt}\right)$$

In this expression, n is the number of electrons involved in the reaction, F is the Faraday constant and $k_{C}$ (a.u. mol^−1^) is a measure of electrochromic efficiency for each channel and for each process.

The first observation is that both the red and green intensity derivatives show peaks similar to the voltammetric peaks, although separated in potential. The peak associated with green appears at more positive potentials, while that associated with red appears at more negative potentials. The intensity derivative of blue shows a variation with ups and downs, which could indicate that it is associated with some intermediate species that appear and disappear during the redox process [13,30].

If we focus on the red and green components, the separation of peak potentials suggests that they may be associated with different processes. Based on these observations, it can be stated that the green color-associated process refers to the electrochemical reactions of the new active centers (bridges between monomer units), and the red color to the centers associated with the phenothiazine ring. This is in good agreement with Figure 4, where a new system of peaks grows during the polymerization (the green color peak) at slightly more positive potentials than the phenothiazine ring peak. In this last case, please note that the phenothiazine ring after reduction loses its planarity and color changes to a less intense blue color, which, in terms of RGB coordinates, indicates an increase in the red component, as observed in [30].

As mentioned above, the use of coupled techniques allows for complex signals to be deconvoluted into their individual components. In this case, and according to Equation (1), it is possible to estimate the contribution to the overall current associated with each color intensity derivative. For this, it is necessary to know the electrochromic efficiency of each color component, or, in other words, the value of $k_{C}$ for each color. As this is not possible a priori, we use a least-squares fitting procedure to fit the sum of the contributions to the experimental current according to Equation (2):(2)$$i=\frac{nF}{k_{R}}\left(\frac{dI_{R}}{dt}\right)+\frac{nF}{k_{G}}\left(\frac{dI_{G}}{dt}\right)$$

This fitting was conducted for data in the potential range corresponding to the cathodic peak. Electrochromic efficiencies obtained from the fitting for red and for green are $\frac{k_{R}}{n}=3.4\times 10^{9}\;\left(a.u.{\;\mathrm{mol}}^{-1}\right)\;$ and $\frac{k_{G}}{n}=0.83\times 10^{9}\;a.u.{\;\mathrm{mol}}^{-1}$. These are relatively high electrochromic efficiencies, particularly for red, and they are similar to values obtained for poly(o-toluidine) [50]. Figure 7 shows the deconvolution of the electrochemical current response during the voltammogram as the sum of two processes, one associated with the green color change and the other with the red color change.

It is also interesting to note that although the red color change is much more evident, the current associated with this process is much lower than that associated with green. In addition, Figure 6 and Figure 7 show that the color intensity derivatives have baseline = 0. It is especially observed at more negative potentials, where the current starts to decrease due to the hydrogen evolution reaction (HER). This is why the difference between the experimental current and the sum of the two contributions is especially distinct in this region of the voltammogram. In addition, during the anodic peak, it is observed that the parameters obtained for the cathodic peak do not reproduce the oxidation peak well. The line marking the difference shows discrepancies. Therefore, we consider that the mechanisms by which reduction takes place are not necessarily the same as for oxidation. A typical way to characterize an electrochromic device is contrast ($\Delta I_{C})$, i.e., the difference between maximum and minimum color intensity, as given by Equation (3).
(3)$$\Delta I_{C}=I_{C,max}-I_{C,min}$$

Figure 8 shows the evolution of contrast with the scan rate. In this case, and for a scan rate of 2 mV s^−1^, we obtain a contrast of 51 units for the red component and 27 units for the green component. Assuming that all the active centers have been reduced, that means a surface concentration for the red active centers of 0.015 ${\mu \mathrm{mol}\;\mathrm{cm}}^{-2}$ and 0.042 ${\mu \mathrm{mol}\;\mathrm{cm}}^{-2}$ for the green active centers. In Figure 4, it is shown that the system of peaks associated with the phenothiazine ring proves smaller than the system of peaks associated with the new bridges between the monomer units. It is compatible with these surface concentrations. The green color processes could be identified with the electrochemical activity of the new bridges and the red color processes with the activity of the phenothiazine ring. Note that a pure blue color surface means a red color intensity of 0, and then, if the polymer blanches during reduction, the $I_{R}\;$ component should increase. This is not so visible for $I_{B}$, since both the reduced and the oxidized forms are blue. According to the electropolymerization mechanism proposed by Liu and Mu [40], there are two N for each new generated bridge between monomers and there are two bridges for each monomer unit, while there is only one phenothiazine ring. This makes the number of active centers associated with the bridges in the polymer significantly higher than those associated with the phenothiazine ring, in good agreement with these previous results.

Figure 8 shows that contrast for the red component proves more dependent on the scan rate. This fact can be interpreted as the phenothiazine ring’s electroactivity being slower than the electroactivity of the bridges of this monomer.

### 2.4. Scan Rate Study of Voltammetric Peaks

We present the study of the dependence of the voltammetric curves on the scan rate. The most used model to describe the voltammetric curves of conductive polymers deposited on electrodes is the classical model, in which electroactive substances (oxidized and reduced) are adsorbed on the electrode surface. In this case, symmetrical voltammetric peaks are obtained; ideally, there is no separation between the cathodic and anodic peak and the height of the peaks is similar. When the scan rate is changed, the peak intensity increases linearly with the scan rate accordingly to the Laviron model [47]. Since it seems possible to identify each color derivative with a different electrochemical (electroactive center) process, we study the dependence of the electrochemical current peak ($i_{p})$ and the maximum of color derivatives for the red and green components, ${\left(\frac{dI_{R}}{dt}\right)}_{p}$ and ${\left(\frac{dI_{G}}{dt}\right)}_{p}$, respectively (Figure 9).

It is observed that while the cathodic peak currents and the maxima of the color intensity derivative vary linearly with the scan rate, for the anodic process, the variation is linear with the square root of the scan rate. For the cathodic process, the rate control is given by the kinetics of the process, while for the anodic process, it appears to be given by transport control. To understand this point, it must be considered that the transport control can be due to the transport of electroactive species to or from the solution, but also the transport of electrons in the conducting polymer. Once reduced, the phenothiazine ring of the polymer loses its structural planarity by the reduction of some double bonds and this means that it can also lose part of its conductive properties by losing part of the conjugated double bond structure [13,15]. In other words, this transport control may also indicate that the polymer is less conductive in its reduced form.

### 2.5. pH Dependence of Voltammetric Peaks

Finally, we study the effect of pH on the mechanism by which electrochemical reactions take place from the measurement of transmitted light. Figure 10 shows the voltammograms obtained at pH = 1.02, pH = 3.12 and pH = 5.72 at a rate of 10 mV s^−1^ and in a 0.5 M KNO_3_ solution. First, it should be noted that the potential window is different in the three cases. This is because the electrochemical response of this system is highly dependent on the pH of the solution. It is already known that the reduction of the monomer requires the protonation of different centers in the ring and terminal amino groups [51].

Looking at the shape of voltammograms at different pH, one can see that the HER takes place together with the electrochemical reduction of the PMB at pH = 5.72, while it is clearly separated in the voltammogram at pH = 1.02. While the HER is shifted by 59 mV/pH at room temperature, the redox processes associated with the polymer are shifted more significantly. A first estimate is obtained by plotting the variation in the peak potentials for the current and the color intensity derivatives (Figure 11). The fittings show a higher correlation for the peak potentials obtained from the color intensity derivatives than for the overall current. This occurs since the current is the sum of more than one contribution and the color intensity derivatives are associated with simpler redox processes. On the other hand, it is obtained that the potential of the peak of the $dI_{G}/dt$ moves at about 120 mV/pH, while that of the peak of $dI_{R}/dt$ does so at about 160 mV/pH.

To further interpret these results, we will consider the proposed reaction mechanisms for monomer reduction. Leventis and Chen proposed that MB gains 3H^+^ during the reduction of the phenothiazine ring involving two electrons [51]. Figure 12 shows the proposed reaction at pH < 6. In this case, reduction implies a color change from intense blue to the reduced colorless form.

If we compare these results with the case of PMB, and according to what we had discussed above, we can assign the process of reduction of the phenothiazine ring to the changes in red color intensity. If we focus on the green color changes that we have associated with the electroactive processes of the bridges between monomeric units, we observe a shift at 120 mV/pH, which means that the reduction mechanism involves the insertion of two protons before the first electron transfer [34]. Since there are two amino groups, these two protons can go to the amino groups. After this protonation, there is the electrochemical reduction process associated with the bridges. However, this is not the reaction causing most of the color change in the PMB. The red color-associated reduction reaction which takes place at more negative potentials occurs after and, according to the mechanism proposed for the MB reduction, it needs the amino groups to be protonated. However, this reaction only needs one more proton to be fixed on the nitrogen of the phenothiazine ring.

Then, a mechanism for the reduction process compatible with those results could include a first step of protonation (2H^+^) and then a transfer of two electrons (Figure 13). This process is associated with green color changes:

A second step in this mechanism will be the protonation of the nitrogen in the phenothiazine ring and then the electrochemical reduction of the structure, which causes the disappearance of the characteristic blue color. Since this second step needs the previous protonation of the amino groups, apparently 3H^+^ are needed before the first electron transfer. This mechanism also explains the 160 mV/pH dependence of the red color peak intensity.

Concerning the oxidation process, it is observed that the electrochromic efficiency values obtained in the cathodic peak zone do not allow for such an accurate simulation of the anodic current. The latter, together with the fact that the anodic peak intensities vary with the scan rate, with a lower slope than the cathodic peak intensities, and with the fact that the anodic peak generally appears to be smaller and wider, allows us to assume that the oxidation mechanism is different from the reduction mechanism. Therefore, this will require further investigation.

## 3. Materials and Methods

All chemical products were commercial. HCl 37%, reagent-grade, ACS, ISO (Scharlau, Spain); KOH (Merck, for analysis, Darmstadt, Germany); methylene blue (Fluka, Seelze, Germany); KNO3 (Emsure, for analysis, Darmstadt, Germany); HNO_3_ 60%, extra pure (Scharlau, Barcelona, Spain); Na_2_C_2_O_4_, extra pure (Panreac, Barcelona, Spain); EtOH absolute, spectroscopy-grade (Spectrosol, Barcelona, Spain); CH_3_COOH, synthesis-grade (Scharlau, Barcelona, Spain). Water was deionized (MilliQ-plus, Millipore, resistivity 18.2 MΩ cm, Darmstadt, Germany).

For the MB solution, firstly, 0.0077 g of MB was weighed; to this, 1 mL of absolute ethanol and 0.4 mL of acetic acid were added, and this was taken to a 0.01 M solution of Na_2_C_2_O_4_ in water. Then, the pH of the solution was adjusted by adding previously prepared KOH or HCl aqueous solutions. Two different solutions were tested: an acid solution (pH 1.67) with 0.1 M HCl and a basic solution (pH = 11.79) with 0.1 M KOH. Deposition was carried out by cyclic voltammetry at a potential range from −0.4 V to 1.0 V at a scan rate of 20 mV s^−1^ during 100 cycles [52]. The SS316 electrodes were mechanically polished before the deposition to remove any remaining passivate layers or other unwanted substances, then washed with ethanol and deionized water.

The characterization studies were carried out in 0.5 M KNO_3_ aqueous solutions, where pH was adjusted by adding 0.1 M KOH or 0.1 M HNO_3_ solutions until the desired pH was reached.

pH was measured with a pH Meter, 0.00–14.00 pH Measurement Range pH Tester, Digital pH Meter. Characterization studies at three different pH were performed (pH = 1.02, pH = 3.12 and pH = 5.72). At pH = 3.12, the effect of the scan rate on the peak currents and peak derivatives of color intensities was achieved (2 mV s^−1^ to 200 mV s^−1^) in the potential range [−0.4;0.5] V. At pH = 1.02, the voltammogram was obtained at 10 mV s^−1^ in the potential range [−0.6;0.7] V, and at pH = 5.72, the voltammogram was obtained at 10 mV s^−1^ in the potential range [−0.8;0.5] V.

All the electrochemical experiments were carried out in a typical three-electrode cell, where the reference was Ag|AgCl, the auxiliary was a Pt mesh and the working electrode was a 6 × 1 cm SS316 sheet covered with Teflon tape, leaving an uncovered surface of 1 × 1 cm at the end of the sheet. The electrochemical cell was an optical-quality glass cell of 4 cm × 2 cm. During the characterization experiments, a HUE HD Pro Camera, Model: PC0000 (S0011) was attached to the cell, allowing for video-recording at 30 frames/s with 8 bit depth and 1280 × 720 pixel resolution. A Potentiostat-Galvanostat μStat400 was used for the electrogeneration studies and AUTOLAB PGSTAT 302 for the characterization experiments.

All the electrochemical characterization experiments were replicated and two voltammetric cycles were recorded at each scan rate and pH. No significant differences were observed among the different voltammetric series and cycles.

Characterization experiments were conducted in a home-made white box illuminated by a LED lamp (6500 K, 10 W). The light exposure of the camera was fixed at the start of the experiments to ensure better reproducibility. Figure 14 shows the experimental setup of this spectroelectrochemical cell.

The acquired video was analyzed with the help of home-made software, which was made for this purpose. Each video was separated into individual images from which the color evolution of the region corresponding to the electrode surface was analyzed. For a better understanding of the results, the mean color intensities for the three channels (${\bar{I}}_{C}$) of a digital image (red, green and blue) were obtained for each frame from Equation (4), thus obtaining the temporal evolution of these parameters.
(4)$${\bar{I}}_{C}=\sum_{j=1}^{n_{p}}\frac{I_{C}\left(j\right)}{n_{p}}$$
where $n_{p}$ is the number of pixels of the analyzed region of the digital image.

The analysis of the digital images was carried out with software programmed in Lazarus 3.2 (RGB_extractor© 2.0 from the Electrochemistry Laboratory of the University of Valencia, Spain).

## 4. Conclusions

Methylene blue can be polymerized on the surface of stainless steel 316 electrodes if prepared in a medium that protects the monomer from possible degradation due to the presence of reactive oxygen species at potentials where the radical cation monomer formation takes place. This medium must be sufficiently basic, and the addition of oxalate anions allows for the stabilization of the monomer.

The polymer shows electrochromic activity that is highly dependent on pH. Video electrochemical analysis (DVEC) shows that the electroactive process associated with the phenothiazine ring is associated with red color intensity changes, while the electroactive processes associated with the new bridges between monomeric units are associated with green color intensity changes.

The shift of peak potentials with pH allows us to postulate a possible mechanism for the reduction process in which two protons enter the polymer before the first electron transfer associated with the green color intensity processes and a third proton enters the film before the reduction process of the phenothiazine ring associated with the red color changes.

Finally, and as a methodological conclusion, it is stated that the use of video electrochemistry together with electrochemical techniques provides valuable information on mechanistic aspects of complex electrochromic processes.

## Figures and Tables

**Figure 1 molecules-29-03752-f001:**
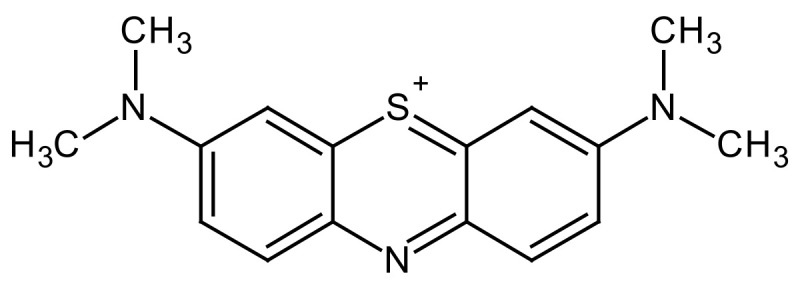
Chemical structure of methylene blue.

**Figure 2 molecules-29-03752-f002:**
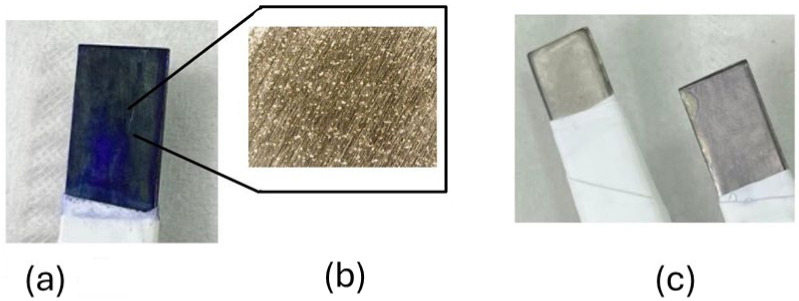
(**a**) Methylene blue deposit in acidic medium with pitting formation, showing very intense blue color. (**b**) Magnification of part of electrode image in (**a**) where pits on the electrode surface are visible. (**c**) Electrode after characterization in 0.5 M KCl that has lost color (**right**), compared to unused SS316 electrode (**left**).

**Figure 3 molecules-29-03752-f003:**
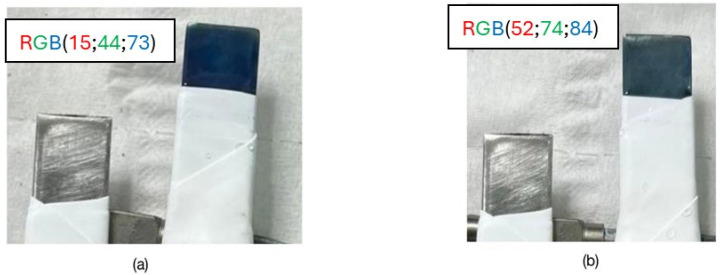
(**a**) Freshly electrogenerated PMB on SS316 in basic medium, free of Cl^−^ anions and with $C_{2}O_{4}^{2-}$; (**b**) PMB on SS316 after characterization in 0.5 M KNO_3_ solution. RGB coordinates of each blue electrode are marked.

**Figure 4 molecules-29-03752-f004:**
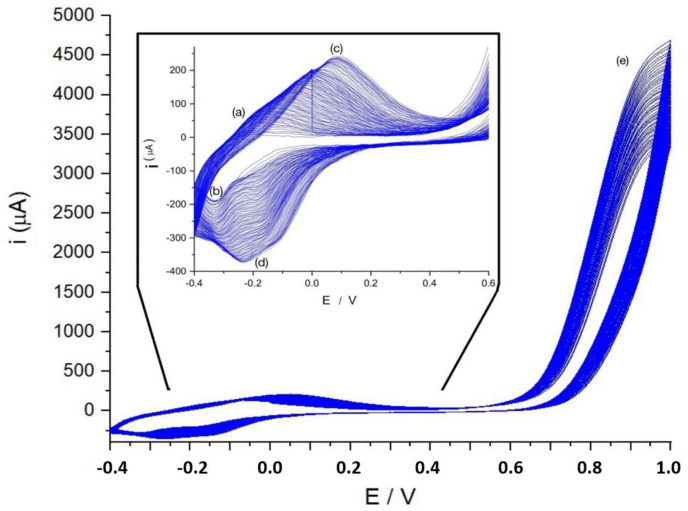
Polymerization of MB by cyclic voltammetry at potentials ranging from −0.4 V to 1.0 V, scan rate 20 Mv s^−1^. In this Figure: (a) anodic peak of MB at −0.22 V; (b) cathodic peak of MB at −0.33 V; (c) anodic peak of PMB at 0.08 V; (d) cathodic peak of PMB −0.23 V; (e) potentials where radical cation is formed and polymerization takes place.

**Figure 5 molecules-29-03752-f005:**
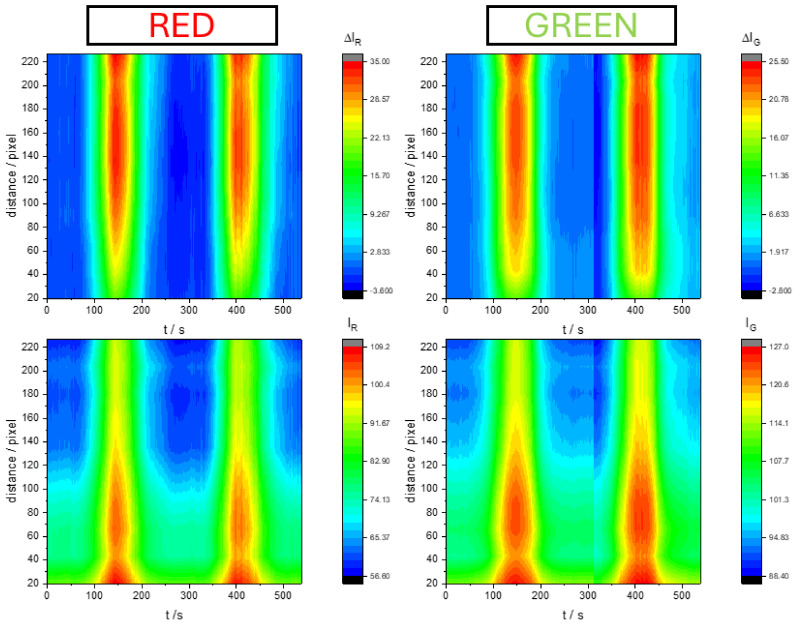
Three-dimensional map of color intensity increment ($\Delta I_{R},\;\Delta I_{G})$ or color intensity ($I_{R}$,$I_{G}$) against the distance (in pixel) to the top of the electrode and the elapsed time during the voltammogram of PMB on SS316 electrodes in 0.5 M KNO_3_ pH = 3.12 solution. The scan rate was 10 mV s^−1^. The electrode length was 1 cm; 0 pixels correspond to the end of Teflon tape and 230 pixels correspond approximately to 1 cm from the Teflon tape (see Figure 3).

**Figure 6 molecules-29-03752-f006:**
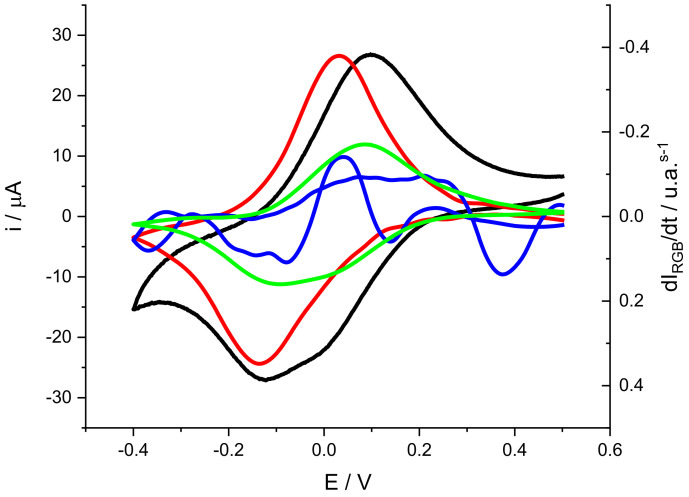
Voltammogram of a PMB polymer generated on SS316 (black line). Experimental conditions: 0.5 M KNO_3_ solution, pH = 3.12, scan rate = 0.002 V s^−1^. Color changes are expressed as mean color intensity derivatives for the red (red line), green (green line) and blue (blue line) components.

**Figure 7 molecules-29-03752-f007:**
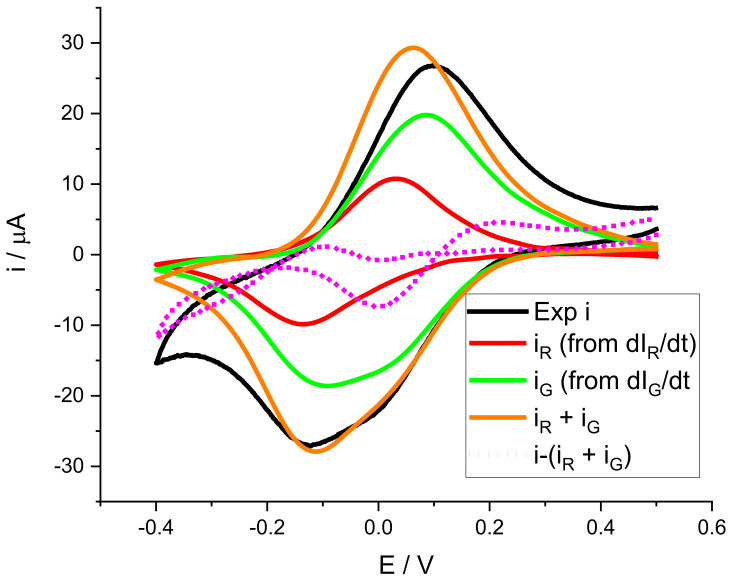
Deconvolution of a voltammogram of a PMB polymer generated on SS316. Experimental conditions: 0.5 M KNO_3_ solution, pH = 3.12, scan rate = 0.002 V s^−1^. The black line corresponds to the experimentally measured current during the experiment; the red line and the green line represent the current associated with the $dI_{R}/dt$ and $dI_{G}/dt$ curves of Figure 6. Dotted lines represent the differences between the experimental current and the contributions of red and green processes.

**Figure 8 molecules-29-03752-f008:**
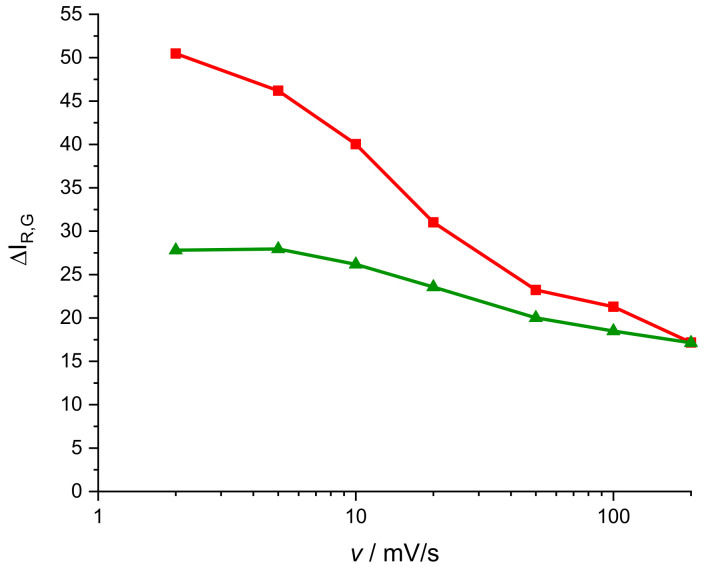
Representation of the contrast of red and green color intensities $\Delta I_{C}\;$(Equation (3)) versus the scan rate (mV s^−1^) during the CV of the PMB process in a 0.5 M KNO_3_ solution, pH = 3.12. Red lines correspond to the red color contrast and green line to the green color contrast.

**Figure 9 molecules-29-03752-f009:**
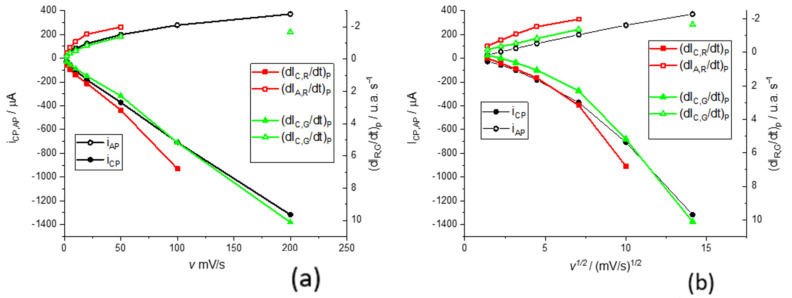
Anodic and cathodic peak currents and the maximum of color derivatives for the red and green components against the scan rate (**a**) and against the square root of the scan rate (**b**). In this context, i is the electrical current and dI_C_/dt the color intensity derivative for color C. Subindex C and A make reference to the cathodic and anodic processes, respectively. All experiments were carried out in a 0.5 M KNO_3_ pH = 3.12 at different scan rates. Open symbols are for anodic peaks and filled symbols for cathodic peaks.

**Figure 10 molecules-29-03752-f010:**
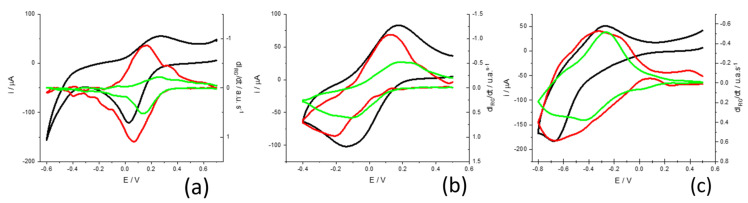
Cyclic voltammograms with color changes in PMB films in 0.5 M KNO_3_ solutions at different pH. (**a**) pH 1.02, (**b**) pH = 3.12, (**c**) pH = 5.72. The scan rate was always 10 mV s^−1^. The black line corresponds to the current curve; red and green lines correspond to the $dI_{R}/dt\;$ and $dI_{G}/dt$ curves, respetively.

**Figure 11 molecules-29-03752-f011:**
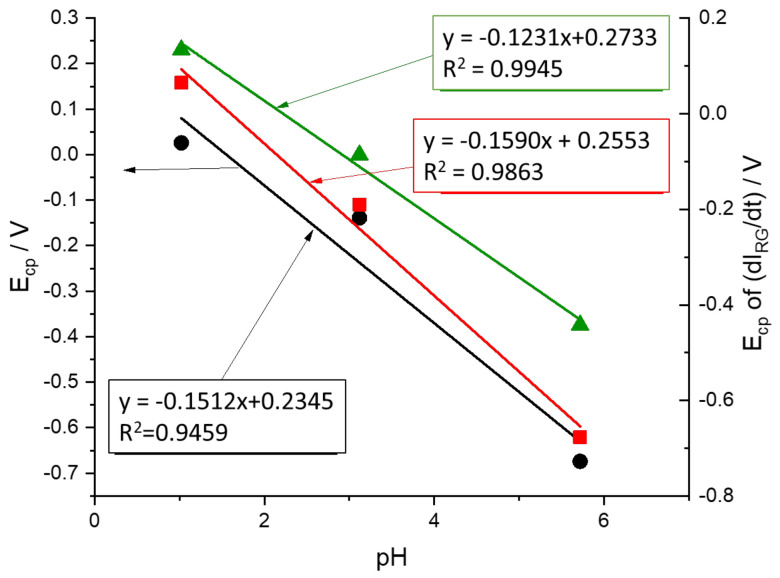
Dependence of cathodic peak potentials on the pH for the current (black line) and the red and green color intensity derivatives (red and green lines, respectively). Scan rate = 10 mV s^−1^; 0.5 M KNO_3_ solution.

**Figure 12 molecules-29-03752-f012:**
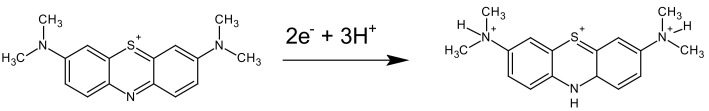
Reduction reaction schema for methylene blue reduction if pH < 6 [51].

**Figure 13 molecules-29-03752-f013:**
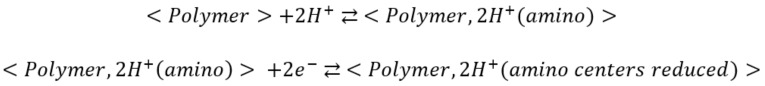
Reduction reaction schema for methylene blue reduction if pH < 6.

**Figure 14 molecules-29-03752-f014:**
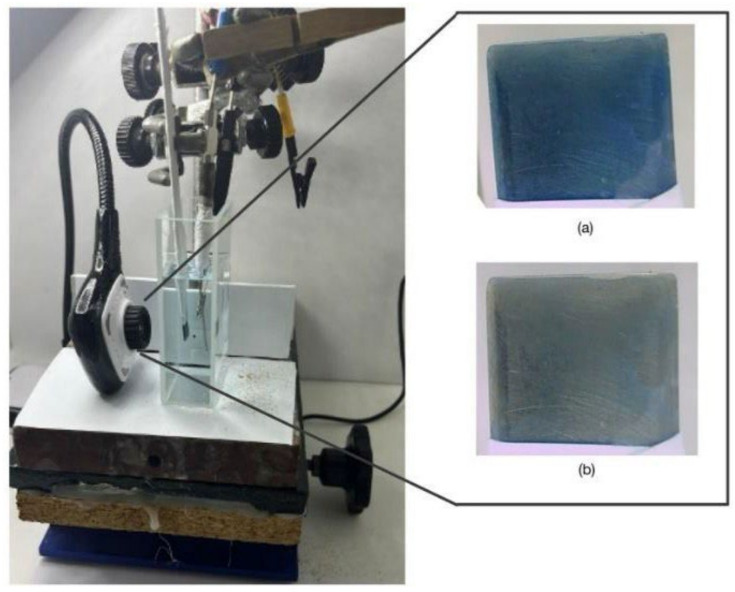
Experimental setup for video acquisition during electrochemical experiments. (a) oxidized PMB on SS316 freshly generated, (b) reduced PMB on SS316.

## Data Availability

Data are available on request.

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
