# Peer review of "Electrogeneration and Characterization of Poly(methylene blue) Thin Films on Stainless Steel 316 Electrodes—Effect of pH"

_molecules, 2024, doi:10.3390/molecules29163752_

Round 1

Reviewer 1 Report

Comments and Suggestions for Authors

The proposed work is very interesting. However, there are some minor points to be changed and a more important change in the structure of the paper.

Minor points are:

1) In figure 2: is not clear what is the golden square in the middle (seems to be a magnification, right?). Please specify at least in the caption of the figure.

2) Figure 5, and the related comment, is not very clear for me. How the distance from the top of the electrode is considered? For example, when this distance is 220 pixel, what is the corresponding part of the electrode? Maybe, it could be useful to add a picture of the electrode indicating the correspondence with the plots of figure 5.

3) About the inhomogeneity of the deposited polymer with the distance, the authors state that can be due to the distribution of the active site, but did you consider problems related the distribution of the electric field in the polymer since there are inhomogeneity in its thickness?   

4) Caption of figure 11 is in Spanish. Please change it.

 About the structure of the paper, I think it could be easier to read if the discussion is merged with the presentation of the results. Actually, the results section contains data but also their discussion. So, I suggest making "Results and discussion" as whole section. 

Author Response

Comments: 1 The proposed work is very interesting. However, there are some minor points to be changed and a more important change in the structure of the paper.

Response 1: First of all, thank you very much for your time and effort in revising our manuscript.

Comments 2: Minor points are:

  • In figure 2: is not clear what is the golden square in the middle (seems to be a magnification, right?). Please specify at least in the caption of the figure.

Response 2: We modified the Figure legend explaining that the golden square was a magnification where the pitting corrosion was observed.

Comments 3:

  • Figure 5, and the related comment, is not very clear for me. How the distance from the top of the electrode is considered? For example, when this distance is 220 pixel, what is the corresponding part of the electrode? Maybe, it could be useful to add a picture of the electrode indicating the correspondence with the plots of figure 5.

Response 3:

Thanks for this comment, it was indeed not correctly explained what this distance was. Distance was measured in pixels from the Teflon tape that covers a part of SS316 electrode (this was 0 pixel) and the end of the electrode was 230 pixel. Since the electrode was 1 cm length, 230 pixel means 1 cm from the top of the uncovered electrode. We have changed the Figure Legend, we have not introduced a photo of the electrode since there are earlier photos that can be referred to (i.e Figure 3).

Comments 4:

  • About the inhomogeneity of the deposited polymer with the distance, the authors state that can be due to the distribution of the active site, but did you consider problems related the distribution of the electric field in the polymer since there are inhomogeneity in its thickness?   

Response 4:

We agree with this idea, we have clarified a little the paragraph where we explain this non-uniformity of color change. In the new version of the manuscript, the paragraph is as follows

In any case, the color changes do not occur as intensely across the whole electrode, possibly because more PMB has been deposited on top, or in other words, the PMB has grown non-uniformly along the electrode. A larger amount of PMB could be generated on the top of the electrode, near the Teflon tape.

Comments 5:

  • Caption of figure 11 is in Spanish. Please change it.

Response 5:

We have changed, thanks again.

Comments 6; 

About the structure of the paper, I think it could be easier to read if the discussion is merged with the presentation of the results. Actually, the results section contains data but also their discussion. So, I suggest making "Results and discussion" as whole section. 

Response 6:

We have changed. We believe it is a good idea for this manuscript.

Reviewer 2 Report

Comments and Suggestions for Authors

In the article, “Electrogeneration and characterization of poly(methylene blue) 2

thin films on stainless steel electrodes. Effect of the pH.” the authors Jose et al. t. aim to present optimization of the electopolymerization process on stainless steel and verify the redox and chromatic properties of the resulting polymer. The study is relevant and addresses significant challenges in the field of conductive polymers, focusing on methylene blue and its potential applications. However, I think following concerns should be addressed before considering for publication. 1. A detail on reproducibility of the experiments and the control conditions is further needed.

2. More detail on the effect of pH on polymer growth and stability is needed with additional data or references.

3. The analysis of the electrochemical data is somewhat superficial and lacks depth in interpreting the results, therefore, conduct more thorough   analysis of the data, e.g. discuss the significance of the peaks observed in the cyclic voltammetry curves in more detail. Explain how these peaks correlate with the polymerization process and properties of the resulting films.

Comments on the Quality of English Language

1. Revised and simplify some sentences for clarity and readability, e.g. Conductive polymers (CPs) have emerged as an exciting class of materials that are attracting significant interest for their conductive properties that have made it possible to replace metals in some applications or semiconductors of an inorganic nature (line 22-24), After several trial-and-error tests, optimum generation conditions were achieved on the freshly polished electrode surface by performing up to 100 voltammetric cycles in the potential range from -0.4 V to 1.0 V against the reference electrode Ag|AgCl at a sweep speed of 20 mV s-1. The monomer solution is brought to pH near 11 and in the presence of oxalate anions, (see Materials and Methods Section).  (line 182-186). And so on.

2. Provide brief explanations for the technical terms when they are first introduced, e.g. cyclic voltammetry, spectroelectrochemistry, PMB, SS316.

3. The title should specify the type of steel used? Think about it. 

Author Response

Comments 1: In the article, “Electrogeneration and characterization of poly(methylene blue) 2 thin films on stainless steel electrodes. Effect of the pH.” the authors Jose et al. t. aim to present optimization of the electopolymerization process on stainless steel and verify the redox and chromatic properties of the resulting polymer. The study is relevant and addresses significant challenges in the field of conductive polymers, focusing on methylene blue and its potential applications. However, I think following concerns should be addressed before considering for publication. 

Response 1: First of all, thank you very much for your time and effort in revising our manuscript.

Comments 2:

  1. A detail on reproducibility of the experiments and the control conditions is further needed.

Response 2:

We agree with the referee’s opinion. We have introduced in the Materials and Methods a paragraph explaining that “All the characterization electrochemical experiments were replicated and two voltammetric cycles were recorded at each scan rate and pH. No significant differences were observed among the different voltammetric series and cycles”. Digital video electrochemistry has been conducted in a home-made white box illuminated by a LED lamp (6500 K, 10 W). If light exposure of the camera is fixed during the experiment a good reproducibility is reached.

Comments 3:

  1. More detail on the effect of pH on polymer growth and stability is needed with additional data or references.

Response 3:

As can be seen in this work, pH is a key factor that must be controlled to obtain an optimum response of this material. The tests to obtain a polymerization in optimal conditions arise from the review of previous works, although on SS316 it may be somewhat different. We studied the effect of pH up to values no higher than 5.7 as there is overlap with hydrogen evolution. We have introduced new references on this topic in the new version of this manuscript. In page 5 we add this information.

An increase in pH causes the monomer units to deprotonate and decreases their solubility. In a recent study it has been proved that the maximum of adsorption of Methylene Blue on a hydrogel is accomplished at pH 10.0 [37]; on the other hand, if the monomer is protonated, in some cases it may be more difficult to polymerize. [38,39]. Besides, the properties of the electrogenerated polymer are dependent on the generation conditions, including the pH of the solution[40–42].

Comments 4: 

  1. The analysis of the electrochemical data is somewhat superficial and lacks depth in interpreting the results, therefore, conduct more thorough   analysis of the data, e.g. discuss the significance of the peaks observed in the cyclic voltammetry curves in more detail. Explain how these peaks correlate with the polymerization process and properties of the resulting films.

Response 4:

In pages 9 and 10, we add an extensive discussion about why the red color and the green color derivative peaks are associated with the phenothiazine aromatic ring active center and the bonds between monomer units electrochemical active centers, respectively. First, the peak potential agrees with the electrogeneration peaks (peaks (c) and (d) of Figure 4). There is also the estimation of the active centers concentrations which agrees with the previous hypothesis.

 Page 9

Form these observations it can be stated that the green color associated process refers to electrochemical reactions of the new active centers (bridge between monomer units) and the red color to the centers associated to the phenothiazine ring. This is in good agreement with Figure 4, where a new system of peaks grows during the polymerization (the green color peak) at slightly more positive potentials (peaks (c) and (d)) than the phenothiazine ring peak (peaks (a) and (b). In this last case, please note that the phenothiazine ring after reduction loses its planarity and color changes to a less intense blue color which means in terms of RGB coordinates an increase in the Red component, as observed[30].

Page 10

Accordingly with the electropolymerization mechanism proposed by Liu and Mu[40], there are two N by each new generated bridge between monomers and there are two bridges by each monomer unit, while only one phenothiazine ring. This makes the number of active centers associated with the bridges significantly higher in the polymer than those associated with the phenothiazine ring in good agreement with previous these results.

Comments 5:

Comments on the Quality of English Language

  1. Revised and simplify some sentences for clarity and readability, e.g. Conductive polymers (CPs) have emerged as an exciting class of materials that are attracting significant interest for their conductive properties that have made it possible to replace metals in some applications or semiconductors of an inorganic nature (line 22-24), After several trial-and-error tests, optimum generation conditions were achieved on the freshly polished electrode surface by performing up to 100 voltammetric cycles in the potential range from -0.4 V to 1.0 V against the reference electrode Ag|AgCl at a sweep speed of 20 mV s-1. The monomer solution is brought to pH near 11 and in the presence of oxalate anions, (see Materials and Methods Section).  (line 182-186). And so on.

Response 5:

We have revised different sentences along the whole manuscript to simplify our language. Thanks for this suggestion.

Comments 6:

  1. Provide brief explanations for the technical terms when they are first introduced, e.g. cyclic voltammetry, spectroelectrochemistry, PMB, SS316.

Response 6:

We have introduced a brief explanation for spectroelectrochemistry, SS316 and PMB. For Cyclic voltammetry we added two references [33,34] in the new version of the manuscript as a reference for this technique.

Comments 7:

  1. The title should specify the type of steel used? Think about it. 

Response 7:

We agree with the referee’s opinion. We have added the SS316 to the title of the manuscript. Electrochemical properties of this polymer could be largely dependent on the stainless steel type.

Round 2

Reviewer 1 Report

Comments and Suggestions for Authors

The manuscript have been improved, and I think that it can be pubblished in the present form. 

Reviewer 2 Report

Comments and Suggestions for Authors

Thanks for revising and improving the manuscript. The manuscript is acceptable in the present form. 

Comments on the Quality of English Language

The quality of English language is sufficient.